# External validation of the QCovid 2 and 3 risk prediction algorithms for risk of COVID-19 hospitalisation and mortality in adults: a national cohort study in Scotland

Steven Kerr,[1] Tristan Millington ![ORCID],[1] Igor Rudan,[1] Colin McCowan,[2] Holly Tibble ![ORCID],[1] Karen Jeffrey ![ORCID],[1] Adeniyi Francis Fagbamigbe,[3,4] Colin R Simpson,[5] Chris Robertson,[6] Julia Hippisley-Cox,[7] Aziz Sheikh[1]

For numbered affiliations see end of article.

**Correspondence to**
Professor Aziz Sheikh;
aziz.sheikh@ed.ac.uk

## ABSTRACT

**Objective** The QCovid 2 and 3 algorithms are risk prediction tools developed during the second wave of the COVID-19 pandemic that can be used to predict the risk of COVID-19 hospitalisation and mortality, taking vaccination status into account. In this study, we assess their performance in Scotland.

**Methods** We used the Early Pandemic Evaluation and Enhanced Surveillance of COVID-19 national data platform consisting of individual-level data for the population of Scotland (5.4 million residents). Primary care data were linked to reverse-transcription PCR virology testing, hospitalisation and mortality data. We assessed the discrimination and calibration of the QCovid 2 and 3 algorithms in predicting COVID-19 hospitalisations and deaths between 8 December 2020 and 15 June 2021.

**Results** Our validation dataset comprised 465 058 individuals, aged 19–100. We found the following performance metrics (95% CIs) for QCovid 2 and 3: Harrell's C 0.84 (0.82 to 0.86) for hospitalisation, and 0.92 (0.90 to 0.94) for death, observed-expected ratio of 0.24 for hospitalisation and 0.26 for death (ie, both the number of hospitalisations and the number of deaths were overestimated), and a Brier score of 0.0009 (0.00084 to 0.00096) for hospitalisation and 0.00036 (0.00032 to 0.0004) for death.

**Conclusions** We found good discrimination of the QCovid 2 and 3 algorithms in Scotland, although performance was worse in higher age groups. Both the number of hospitalisations and the number of deaths were overestimated.

## INTRODUCTION

In December 2019, a novel cCOVID-19 (SARS-CoV-2) emerged in Wuhan, China. The WHO declared the outbreak a public health emergency of international concern on 30 January 2020, and then a pandemic on 11 March 2020. The index case in Scotland was identified on 1 March 2020. As of 26 March 2023, there have been over 2.1 million

## STRENGTHS AND LIMITATIONS OF THIS STUDY

⇒ We used a population-level dataset to perform an external validation of the QCovid 2 and 3 algorithms.
⇒ We validated these algorithms on the same time interval as in the original derivation study.
⇒ We only used a 10% sample of the population due to computational issues.
⇒ Some data used in the algorithms were missing.

COVID-19 cases in Scotland, with over 17 000 deaths.[1]

Algorithms have been developed to identify people who are at risk of severe COVID-19 outcomes such as hospitalisation and death and guide public health policy.[2–7] The QCovid algorithms are risk-scoring systems that predict the probability of COVID-19 hospitalisation and death. The original QCovid algorithm was commissioned by the chief medical officer (CMO) for England on behalf of the UK Government. QCovid was used by the UK Government to inform policies on shielding and vaccine prioritisation for England,[2] and has been independently and externally validated in England,[3] Scotland[4] and Wales.[5] In Scotland, the original QCovid algorithm was found to perform well, with close correspondence between the observed and predicted risks. The Harrell's C scores for hospitalisations and deaths in males and females were 0.809 and 0.946, respectively, over the first period of testing (1 March 2020–30 April 2020).[4] England[3] and Wales[5] had similar results, finding very good performance.

QCovid was updated at the request of the CMO for England to take account of the evolving nature of the pandemic and vaccination status.[6] This consisted of two algorithms:

QCovid 2 which predicted the probability of hospitalisation and death for unvaccinated individuals, and QCovid 3 which predicted the probability of hospitalisation and death for individuals who had received one or two doses of either the ChAdOx1 nCov-19 (Oxford AstraZeneca) or BNT162b2 (Pfizer-BioNTech) vaccines. These algorithms were developed using the QResearch database.[7]

The training cohort for QCovid2 consisted of unvaccinated individuals aged 19–100 years, observed between 1 September 2020 and 31 May 2021. The training cohort for QCovid3 consisted of vaccinated individuals aged 19–100 years old, observed between 8 December 2020 and 15 June 2021.

Following a request from the Scottish Government, we sought to externally validate these QCovid2 and 3 algorithms for the adult Scottish population. We used a common protocol for validating these algorithms across the four UK nations.[8] We studied the performance of QCovid 2 and 3 between 8 December 2020 and 15 June 2021, the same time period that the algorithm was trained on.

## METHODS
The following information largely overlaps with our previous report[4] validating the first QCovid algorithm, because the methodology was very similar.

### Study design
We carried out an external validation of the QCovid 2 and 3 algorithms using the Early Pandemic Evaluation and Enhanced Surveillance of COVID-19 (EAVE II) platform, which contains electronic health records for 5.4 million (~99% of the population) people in Scotland. QCovid 3 accounts for competing risks, whereas QCovid and QCovid 2 do not. Calculating absolute risk of the outcome of interest in a Cox model with competing risks comes with a significant increase in computational demand. Due to limitations on computational resources, we performed this validation using a random 10% sample of the cohort.

### Datasets
We had data from all 940 Scottish primary care practices. These were linked to the Electronic Communication of Surveillance in Scotland (national database for all virology testing including NHS and UK Government test centre data), the Scottish Morbidity Record (record of hospitalisation data) and National Records Scotland (death certification) data as part of the EAVE II platform. A more detailed description of the data can be found in our cohort profile in online supplemental material. Online supplemental table S1 shows a description of the cohort as well as the 10% sample. A data dictionary for the EAVE II project is available online.[9]

### Selection criteria
Any individual in the linked dataset aged between 19 and 100 years on 8 December 2020 was eligible for inclusion.

Anyone who had a COVID-19 related hospitalisation before the start of follow-up for each dose was excluded from the hospitalisation analysis.

### Exposures
Predictor variables were those used in the QCovid 2 and 3 algorithms.[6] These are detailed in online supplemental box 1. All predictor variables were taken as the most recent recorded value in the data at cohort entry.

### Outcomes
The primary outcomes were time to COVID-19 hospitalisation, and time to COVID-19 death. COVID-19 hospitalisation was defined as hospitalisation with a positive reverse-transcription PCR (RT-PCR) COVID-19 test within 28 days prior to or admission with ICD-10 (International Classification of Disease version 10) codes for COVID-19 (U07.1, U07.2). COVID-19 death was defined as all-cause death within 28 days of a post-positive RT-PCR test, or death with ICD-10 codes for COVID-19 on the death certificate from National Records Scotland.

### Missing data
We could not obtain data relating to some conditions used in the QCovid 2 and 3 algorithms. We did not have reliable data available indicating whether the individual had a bone marrow or stem cell transplant in the last 6 months, whether they had received radiotherapy in the last 6 months, whether they had been prescribed immunosuppressants, oral steroids or antileukotriene or long acting beta2-agonists four or more times in the last 6 months, whether they had irritable bowel syndrome, and whether they had received a solid organ transplant. The values of these variables were set to 'none'. We had data on the type of diabetes, but not if it was controlled. If the individual had type 1 diabetes, we took them to be in the uncontrolled category, and if they had type 2 diabetes, we took them to be in the controlled category, as these were the most populous categories in the training data.[6] For all other comorbidities/treatments, a missing value was taken to indicate absence of that comorbidity/treatment.

Reliable ethnicity data were not available, and all individuals were assigned to 'white British'. In the 2011 Scottish census, 96% of the population was in this ethnic group.[10] The most fine-grained residential location information available in our dataset was data zone, which is a geographical designation comprising groups of UK Census output areas. Output areas typically consist of ~300 people, whereas data zones typically consist of 500–1000 people.[11] Townsend Deprivation Scores[12] for each output area were obtained from the 2011 UK census.[11] We took the median value of Townsend Deprivation Scores for the output areas comprising each data zone to get a deprivation score for each data zone. Missing values for Townsend Deprivation Scores were replaced with the mean value for the cohort. Missing values in the housing category variable were taken to indicate the individual was neither homeless, nor resident in a care home.

We used single imputation by chained equations[12] to impute missing values for body mass index (BMI). There is some evidence of an association between lower levels of socioeconomic status and higher BMI in economically developed countries.[13] Sex is known to be associated with BMI, as is coronary artery disease and diabetes.[14]

There were no missing values for any of the other independent variables.

## Model validation

To calculate the risk for those who were unvaccinated, the baseline survival rate was provided on the following days: 0, 30, 60, 90, 183, 210 and 240. We linearly interpolated the logarithm of the baseline survival rate to obtain values for intermediate days. We calculated the probability of an event for an individual either on the day of the event, or the day of censoring.

We applied versions 2 and 3 of the QCovid algorithm to a random 10% sample of our cohort, and computed Harrell's Concordance,[15] the Brier score, Royston's D,[16] $R^2$[17] and observed-expected (OE) ratio for the period 8 December 2020 to 15 June 2021. As well as calculating these metrics for the entire cohort sample, we also calculated them for several subcohorts. These subcohorts consisted of unvaccinated individuals, individuals followed up after first and second dose, and the age ranges, 19–64, 65–79 and 80–100.

Harrell's concordance is a performance metric that characterises the tendency for people with higher hazard rates (HRs) to have earlier events. It takes values between 0 and 1, with 0 indicating poorer performance and 1 indicating better performance. The Brier score is a measure of forecast accuracy that is equal to the mean squared prediction error in the case of a binary outcome variable. It takes values greater than 0, with 0 indicating the best performance. Royston's D is a measure of 'separation' between survival curves. Higher values of Royston's D indicate predicted HRs have more discriminative ability. $R^2$ is a measure of the proportion of variation in survival time explained by the model. It takes values between 0 and 1, with 0 indicating poorer performance and 1 indicating better performance. We calculated $R^2$ for survival models as defined in.[17] The OE ratio is the number of observed events divided by the expected number of events predicted by the model. A value of 1 indicates that the total number of events is exactly correctly predicted, and a value less than 1 indicates that the total number of events predicted was greater than observed. We made plots of observed versus expected risk on the day of event/censoring by vigintiles (20 groups) of predicted HR. We also calculated recalibrated risk by scaling predicted risks by a multiplicative constant so that the expected total number of events predicted was equal to the observed total number of events. We used this recalibrated risk to calculate recalibrated Brier scores and made recalibrated observed versus expected plots.

## Reporting

This study is reported in accordance with the Transparent Reporting of a multivariable prediction model for Individual Prognosis or Diagnosis guidelines.[18]

## Patient and public involvement

The EAVE II Public Advisory Group reviewed the online QCovid tools and the preliminary results from this study. Although overall in favour of this work, they expressed some concern at the lack of ethnicity data for our results, and that the online tools would need significant editing to be accessible to the public.

## RESULTS

In our 10% cohort sample, there were 763 COVID-19 hospitalisations in the validation period, and 393 COVID-19 deaths. Table 1 shows the OE ratio, Harrell's C, Brier score, Royston's D, $R^2$ and recalibrated Brier score with outcomes of COVID-19 hospitalisation and COVID-19 death for the entire cohort sample, as well as subgroups stratified by follow-up after different vaccine doses. Table 2 shows the same statistics, stratified by age group. Figure 1 plots the observed and expected risk of hospitalisation and death by vigintiles of the HR for the entire sample, the subset who were unvaccinated and those during follow-up after first and second dose, and figure 2 shows the same set of figures but recalibrated with the actual number of events. The vigintiles were numbered so that higher vigintiles had a higher HR (ie, those in vigintile 1 were in the lowest HR category, those in vigintile 20 were in the higher HR category).

### Vaccine dose

Focusing first on hospitalisations stratified by vaccine dose (shown in the top half of table 1), according to Harrell's C, Royston's D and the $R^2$ metric, performance was similar across all subgroups, but better for the entire sample cohort. This is because combining data for unvaccinated with vaccinated people created a large number of additional concordant pairs. Both the Brier score and the recalibrated Brier scores were noticeably worse for the unvaccinated than for those who had one or two vaccine doses, or the entire sample cohort.

In terms of COVID-19 deaths (shown in the bottom half of table 1), we found that the model performed similarly according to Harrell's C across the vaccine dose follow-up categories. Again, across the other metrics bar the OE ratio, performance was worse for the unvaccinated than for those who had one or two vaccine doses, or for the entire sample cohort.

For both events, the performance according to Harrell's C was very good and was similar to that in the original study on their validation cohort (0.84 (0.82, 0.86) for hospitalisation in this analysis versus 0.85 in the original study, and 0.92 (0.90–0.94) in this analysis versus 0.93 in the original study).[6] The D statistic performance in this analysis was worse than in the original study (2.21 (2.022,

**Table 1** Performance metrics (with 95% CIs) stratified by vaccination status for COVID-19 hospitalisation and death

| Group | Observed-expected ratio | C | Brier score | Royston's D | Recalibrated Brier score | $R^2$ |
|---|---|---|---|---|---|---|
| Hospitalisation | | | | | | |
| Everyone | 0.24 | 0.84 (0.82 to 0.86) | 0.0009 (0.00084 to 0.00096) | 2.12 (2.022 to 2.218) | 0.00091 (0.00085 to 0.00097) | 0.52 (0.49 to 0.54) |
| Unvaccinated | 0.23 | 0.73 (0.71 to 0.75) | 0.00209 (0.00193 to 0.00225) | 1.33 (1.2124 to 1.4476) | 0.00211 (0.00195 to 0.00227) | 0.3 (0.26 to 0.33) |
| First and second dose vaccinated | 0.33 | 0.72 (0.66 to 0.78) | 0.00015 (0.00013 to 0.00017) | 1.43 (1.136 to 1.724) | 0.00015 (0.00013 to 0.00017) | 0.33 (0.23 to 0.42) |
| Death | | | | | | |
| Total population | 0.26 | 0.92 (0.9 to 0.94) | 0.00036 (0.00032 to 0.004) | 2.77 (2.6132 to 2.9268) | 0.00036 (0.00032 to 0.0004) | 0.65 (0.62 to 0.67) |
| Unvaccinated | 0.34 | 0.88 (0.86 to 0.9) | 0.00077 (0.00069 to 0.00085) | 2.35 (2.1736 to 2.5264) | 0.00077 (0.00069 to 0.00085) | 0.57 (0.53 to 0.6) |
| First and second dose vaccinated | 0.13 | 0.94 (0.92 to 0.96) | 0.00001 (0.00008 to 0.00012) | 3.09 (2.7372 to 3.4428) | 1.0001 (0.00008 to 0.00012) | 0.7 (0.64 to 0.74) |

**Table 2** Performance metrics (with 95% CIs) for COVID-19 hospitalisation and death stratified by age group

| Age group | Observed-expected ratio | C | Brier score | Royston's D | Recalibrated Brier score | $R^2$ |
|---|---|---|---|---|---|---|
| Hospitalisation | | | | | | |
| Age 18–64 | 0.21 | 0.82 (0.8 to 0.84) | 0.00067 (0.00061 to 0.00073) | 1.9 (1.7628 to 2.0372) | 0.00068 (0.00062 to 0.00074) | 0.46 (0.42 to 0.5) |
| Age 65–79 | 0.29 | 0.80 (0.78 to 0.82) | 0.00112 (0.00098 to 0.00126) | 1.70 (1.504 to 1.896) | 0.00113 (0.00099 to 0.00127) | 0.41 (0.35 to 0.46) |
| Age 80+ | 0.27 | 0.78 (0.76 to 0.8) | 0.00255 (0.0022 to 0.0029) | 1.55 (1.3344 to 1.7656) | 0.00258 (0.00223 to 0.00293) | 0.36 (0.3 to 0.43) |
| Death | | | | | | |
| Age 18–64 | 0.15 | 0.85 (0.81 to 0.89) | 0.00007 (0.00005 to 0.00009) | 1.96 (1.5288 to 2.3912) | 0.00007 (0.00005 to 0.00009) | 0.48 (0.36 to 0.58) |
| Age 65–79 | 0.28 | 0.82 (0.8 to 0.84) | 0.00052 (0.00042 to 0.00062) | 1.77 (1.476 to 2.064) | 0.00052 (0.00042 to 0.00062) | 0.43 (0.34 to 0.51) |
| Age 80+ | 0.3 | 0.73 (0.71 to 0.75) | 0.00309 (0.0027 to 0.00348) | 1.23 (1.034 to 1.426) | 0.00311 (0.0027 to 0.00352) | 0.26 (0.2 to 0.33) |

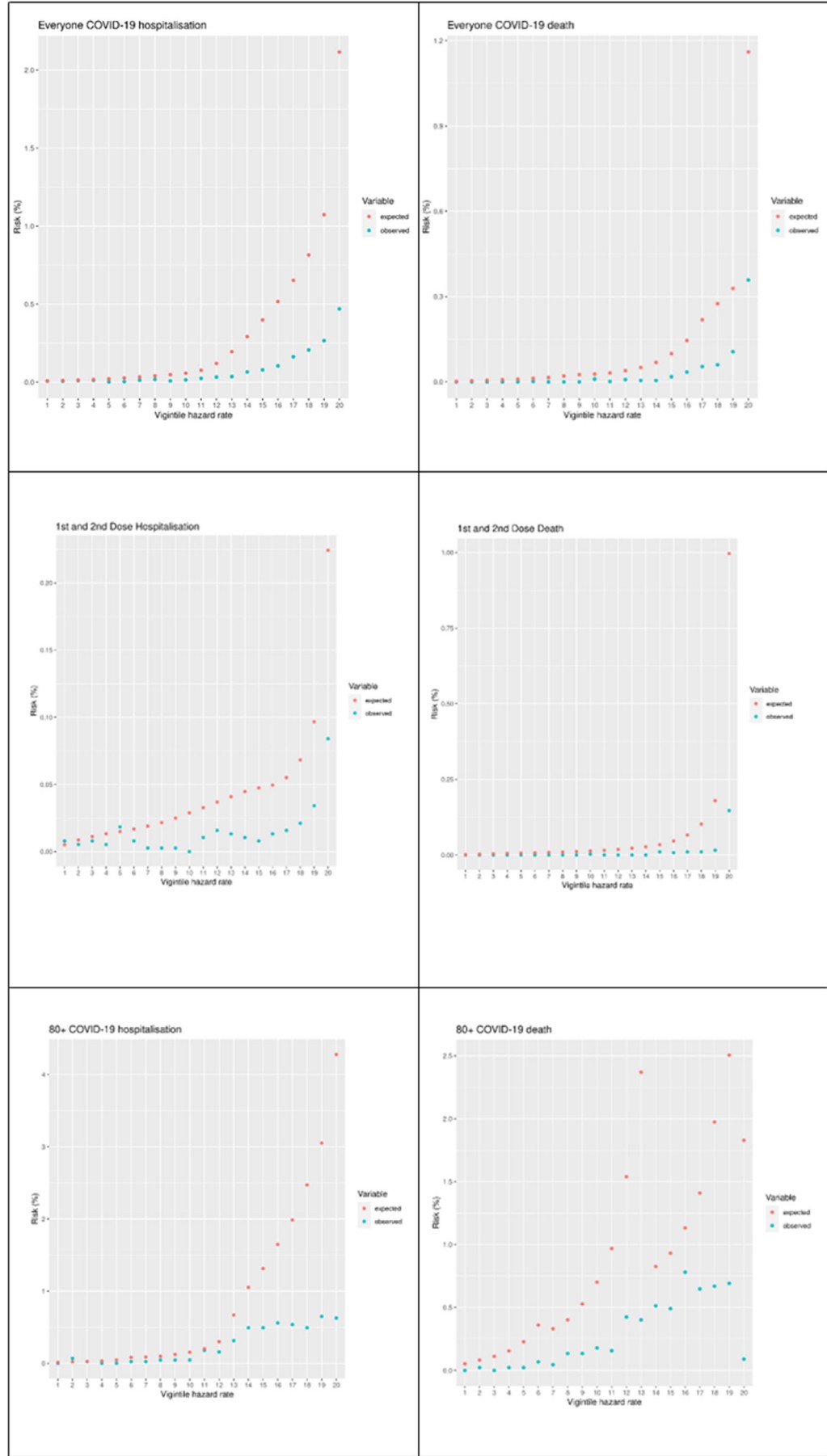

**Figure 1** Vigintile plots of expected versus observed risks for the cohort and a selection of subcohorts.

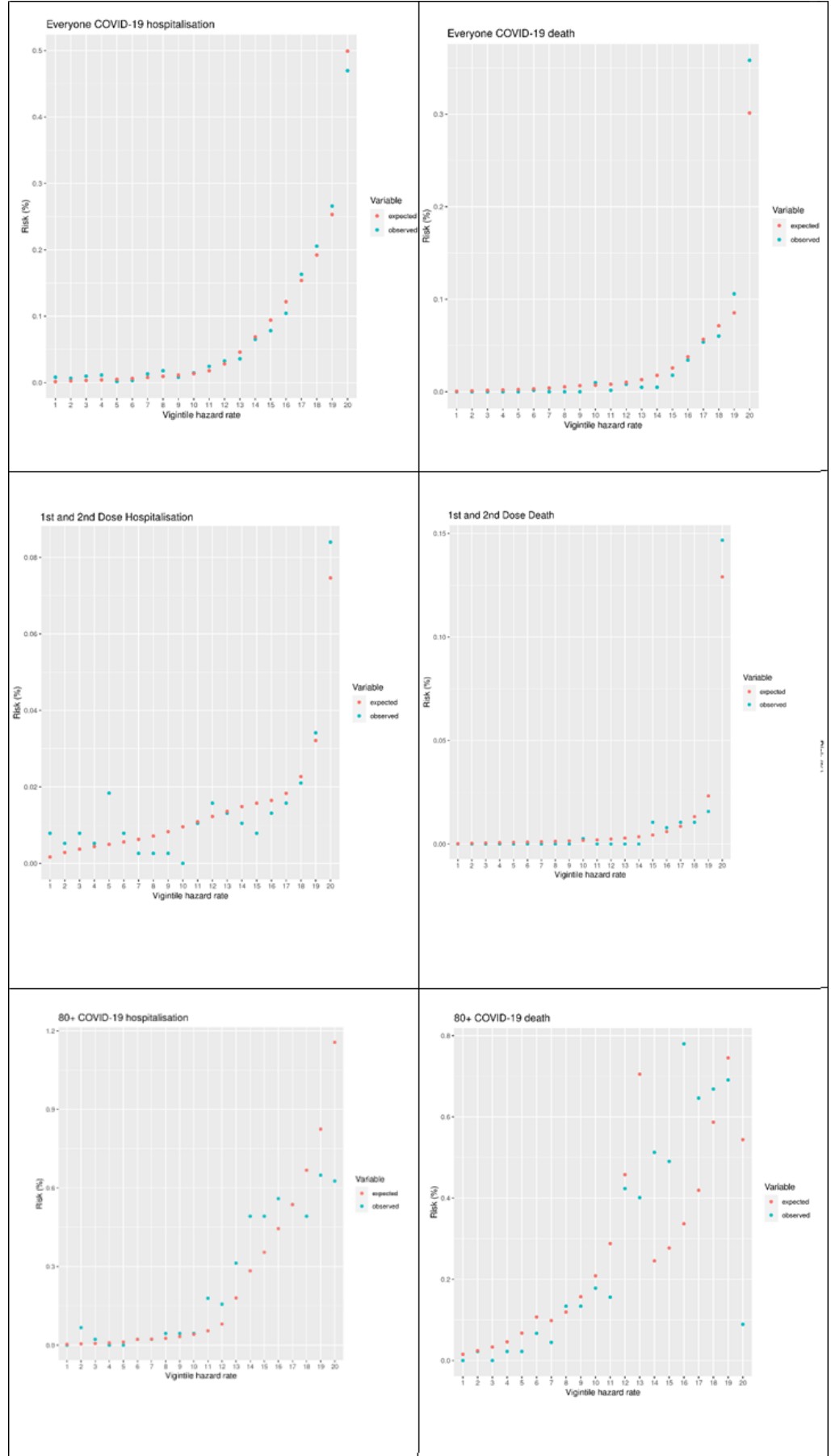

**Figure 2** Vigintile plots of expected versus observed recalibrated risks for the cohort and a selection of subcohorts.

2.218) vs 2.83 (2.59, 3.08) for hospitalisation and 2.77 (2.61, 2.93) vs 3.46 (3.19, 3.73) for death).

## Age
Looking at the results for COVID-19 hospitalisation stratified by age (shown in top half of table 2), the model performed less well for the older age groups (65–79 and ≥80) compared with the younger group (18–64) according to all metrics, except for the OE ratio. This pattern is also repeated for COVID-19 death stratified by age (shown in the bottom half of table 2).

## Vigintile plots
Figure 1 shows a set of vigintile plots comparing the predicting risk to the observed risk for the full cohort and two subcohorts—those who were unvaccinated and those with a first and second dose. Figure 2 shows the same set of vigintile plots, but with the risks recalibrated.

The vigintile plots on the left-hand side of figure 1 show the model's performance in terms of predicting COVID-19 hospitalisations (left-hand side) and deaths (right-hand size). For both events, the models tended to overestimate the events across all stratifications. The estimates tended to be worse for the higher risk vigintiles.

Figure 2 shows the same plots after we recalibrated the models. After this we found that most of the vigintiles had a good estimation of the number of events after recalibration. Across the stratifications, there was no consistent pattern of overestimating or underestimating the number of events for specific vigintiles. The predictions for the ≥80 years subcohort, particularly for death, were not as good as for the other cohorts. This may have been due to the small sample size combined with the larger risk scores.

## DISCUSSION
### Statement of principal findings
In this external validation of the QCovid 2 and 3 algorithms in Scotland, we found the algorithms performed well with regard to discrimination as measured by the C statistic. The total number of events was overestimated for hospitalisations and deaths across all subgroups. Performance was broadly similar to that in the original study.[6]

### Strengths and limitations
Our study had a number of important strengths. As in our previous study,[4] we developed a unique linked dataset covering 99% of the population resident in Scotland. The EAVE II database[19] is one of the few national individual patient-level linked research databases in the world.[20] We evaluated the performance of the QCovid 2 and 3 algorithms according to all metrics used in the original study[6] and in the common protocol agreed between the four UK nations.[8]

However, our work has several limitations. First, we did not have access to ethnicity data, so all individuals were set to 'white British' ethnicity. We believe modal substitution for ethnicity was reasonable because the most

recent Scottish census indicated that 96% of the residents of Scotland identified their ethnicity as 'white'.[10] There was significant missingness in the BMI data, with 2 495 504 (55.6%) missing values. We used single imputation by chained equations to impute these missing values for BMI. Multiple imputation was not used due to limitations on computational resources. The most fine-grained residential location information available in our dataset was data zone, which typically consists of multiple 2011 UK census output areas. We took the median value of the Townsend Deprivation Scores for the output areas comprising each data zone to get a deprivation score for each data zone. Missing values of Townsend Deprivation Scores were replaced with the average value for the cohort. Higher levels of deprivation as measured by Townsend Deprivation Scores were associated with increased predicted risk of COVID-19 hospitalisation and death in the QCovid 2 and 3 algorithms. We also did not have data for several clinical risk variables, so individuals were assigned to the category for absence of the condition. This will have had the effect of under-estimating risk in people with these characteristics. We also validated on a 10% sample randomly chosen rather than the full EAVE II cohort due to limitations on computational resources. This led to there being few samples in some of the higher-risk groups and wider CIs on the metrics.

### Interpretation
The QCovid 2 and 3 risk prediction algorithms performed well in the Scottish population in the period they were trained for.

### Implications for policy, practice and research
Our results indicate that QCovid 2 and 3 would have been appropriate for use as a risk prediction tool for COVID-19 hospitalisation and death in Scotland during our study period.

For future research, taking UK-wide perspectives on data availability when developing risk prediction tools should be considered if these tools are to be used nationally.

## CONCLUSION
Risk prediction tools are valuable for identifying individuals at the highest risk of experiencing severe outcomes and can be used by policy-makers to help mitigate these risks. However, these tools must be externally validated on cohorts that were not used to train them. We evaluated the QCovid2 and 3 algorithms in Scotland using a variety of metrics and across several subgroups. We found good performance overall, and many measures were similar to those in the original study.

**Author affiliations**
[1]Usher Institute of Population Health Sciences and Informatics, The University of Edinburgh, Edinburgh, UK
[2]School of Medicine, University of St. Andrews, St Andrews, UK
[3]Institute of Applied Health Sciences, University of Aberdeen, Aberdeen, UK

[4]Department of Epidemiology and Medical Statistics, University of Ibadan, Ibadan, Nigeria
[5]Faculty of Health, Victoria University of Wellington, Wellington, New Zealand
[6]Department of Mathematics and Statistics, University of Strathclyde, Glasgow, UK
[7]Nuffield Department of Primary Care Health Sciences, University of Oxford, Oxford, UK

**Acknowledgements** The authors would like to thank staff at Public Health Scotland, Albasoft, the general practices that contributed data to the study and the EAVE II Collaborators. AS and CR serve on The Scottish Government's COVID-19 Chief Medical Officer's Advisory Group and the New and Emerging Respiratory Virus Threats Advisory Group (NERVTAG) Risk Stratification Subgroup.

**Contributors** AS and JH-C conceptualised the study. SK carried out the formal analysis. TM wrote an initial draft of the manuscript. All authors assisted with review and editing. CR and TM have verified the underlying data. AS acted as guarantor.

**Funding** National Institute for Health Research (NIHR) following a commission by the Chief Medical Officer for England. EAVE II is funded by the Medical Research Council (MC_PC_19075) with the support of BREATHE: the Health Data Research Hub for Respiratory Health (MC_PC_19004), which is funded through the UK Research and Innovation Industrial Strategy Challenge Fund and delivered through Health Data Research UK. Additional support has been provided through Public Health Scotland and the Community Health and Social Care Directorate of the Scottish Government.

**Competing interests** JH-C reports grants from MRC, grants from Wellcome Trust, grants from NIHR, during the conduct of the study; JH-C is a founder and shareholder of ClinRisk and was its medical director until 31 May 2019. ClinRisk produces open and closed source software to implement clinical risk algorithms (outside this work) into clinical computer systems. JH-C was chair of the NERVTAG risk stratification subgroup and a member of SAGE COVID-19 groups and the NHS group advising on prioritisation of use of monoclonal antibodies in COVID-19 infection. AS reports grants from NIHR, grants from MRC, grants from HDR UK, during the conduct of the study. All other authors report no conflict of interest.

**Patient and public involvement** Patients and/or the public were involved in the design, or conduct, or reporting, or dissemination plans of this research. Refer to the Methods section for further details.

**Patient consent for publication** Not applicable.

**Ethics approval** This study involves human participants and ethical permission for this study was granted by the South East Scotland Research Ethics Committee 02 (12/SS/0201). The Public Benefit and Privacy Panel Committee of Public Health Scotland, approved the linkage and analysis of the deidentified datasets for this project (1920-0279). We had approvals to use anonymised patient data without consent.

**Provenance and peer review** Not commissioned; externally peer reviewed.

**Data availability statement** No data are available. All code, metadata and documentation for this project is publicly available at https://github.com/EAVE-II/QCovid2-3-validation. A data dictionary is available at https://github.com/EAVE-II/EAVE-II-data-dictionary. Most of the data that were used in this study are highly sensitive and will not be made available publicly.

**ORCID iDs**
Tristan Millington http://orcid.org/0000-0002-0260-0515
Holly Tibble http://orcid.org/0000-0001-7169-4087
Karen Jeffrey http://orcid.org/0000-0002-0061-6089

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
