## [Reviewer comments · BMJ Open]

ARTICLE DETAILS

TITLE (PROVISIONAL)	External validation of the QCovid 2 and 3 risk prediction algorithms for risk of COVID-19 hospitalisation and mortality in adults: a national cohort study in Scotland
AUTHORS	Kerr, Steven; Millington, Tristan; Ruden, Igor; McCowan, Colin; Tibble, Holly; Jeffrey, Karen; Fagbamigbe, Adeniyi Francis; Simpson, Colin; Robertson, Chris; Hippisley-Cox, Julia; Sheikh, Aziz

VERSION 1 – REVIEW

REVIEWER	Weber, Griffin Harvard Medical School
REVIEW RETURNED	04-Aug-2023

GENERAL COMMENTS	The QCovid algorithm was developed using national health data from England in 2020 to predict hospitalization and death related to COVID-19. The algorithm was independently validated in England, Scotland, and Wales and worked well in all three. The QCovid 2 and QCovid 3 were developed in 2021 to incorporate vaccination status (QCovid 2 for unvaccinated and QCovid 3 for vaccinated). This study is the validation of QCovid 2 and QCovid 3 in Scotland. It had good results, similar to the prior publications. All the QCovid papers have the same structure and share a lot of the same text. This manuscript has large sections that are word-for-word identical to the QCovid Scotland validation paper (Reference #4). (Many of the authors on all these papers are the same.) Normally, this would raise concerns. However, in this case I think it is a strength. The most unique and important aspect of this work is that it started with a national study that was replicated as closely as possible in other nations, with the original predictive model being externally validated in each nation. By having nearly identical papers for each nation, it makes it easy to compare the results. This could be viewed as one large project, broken into separate papers by nation and QCovid algorithm version. Minor comments: 1) What does “recalibrated” mean in Figure 2? How was this done?2) It would be helpful in the discussion to comment on whether the authors have any hypotheses on why the model overestimated risk so much. In particular, the text states that the way they handled missing data would have “had the effect of under-estimating risk”. As a result, the predicted risk might have been even higher if the data were available. Were the patient
--

	populations different in some way between Scotland and England that could explain this? 3) Why were there missing variables in Scotland? Was the model created for England without considering what data are available in Scotland and Wales? Are there any lessons learned from this study about what variables the UK nations have in common and whether future models should or should not focus on those variables?
--	---

REVIEWER	Gathogo, Esther Cerner Corporation, Population Health Management
REVIEW RETURNED	11-Aug-2023

GENERAL COMMENTS	A well written paper. I have a few points to feedback  1. In the dataset you use, did you exclude any patients that may have opted out of use of their data for research purposes? If not, include process that you used to exclude opt out patients. 2. Why did you make the assumption of Type I diabetes = uncontrolled and Type II = controlled. Is there Scottish data on the incidence or prevalence of controlled/uncontrolled of each diabetes type that you could use instead for imputation? 3. Did the database have Height/Weight that you could use to calculate BMI instead of imputation? 4. Is there data on COVID-19 variants being phase II of COVID-19 pandemic? The variants may explain the differences in rates of hospitalisations or death. 5. The model doesn't accurately predict older people, is it possible to evaluate age adjusted co-morbidities eg Charlson comorbidity index which has been reported as a predictor for mortality rate and severe clinical outcome in the patients with COVID-19. 6. I see COPD in your model, do you have data on infective exacerbation of COPD? COPD is one of the most common reasons for hospitalisation.
---

VERSION 1 – AUTHOR RESPONSE

Reviewer: 1
Dr. Griffin Weber, Harvard Medical School

Comments to the Author:

The QCovid algorithm was developed using national health data from England in 2020 to predict hospitalization and death related to COVID-19. The algorithm was independently validated in England, Scotland, and Wales and worked well in all three. The QCovid 2 and QCovid 3 were developed in 2021 to incorporate vaccination status (QCovid 2 for unvaccinated and QCovid 3 for vaccinated). This study is the validation of QCovid 2 and QCovid 3 in Scotland. It had good results, similar to the prior publications.

All the QCovid papers have the same structure and share a lot of the same text. This manuscript has large sections that are word-for-word identical to the QCovid Scotland validation paper (Reference #4). (Many of the authors on all these papers are the same.) Normally, this would raise concerns. However, in this case I think it is a strength. [NOTE FROM THE EDITORS: Wherever such text re-use is present, please ensure it is clearly signposted for the reader, eg, "As in our previous X study, [REFERENCE]"]

RESPONSE: Since most of our methodology is the same as in the previous paper, we have added the following sentence to signpost this to the start of the methods section (lines 149-150):

'The following information largely overlaps with our previous paper [4] validating the first QCovid algorithm, because the methodology was very similar.'

And in the conclusion (lines 336-337):

'As in our previous study, [4] we developed a unique linked dataset covering 99% of the population resident in Scotland.'

The most unique and important aspect of this work is that it started with a national study that was replicated as closely as possible in other nations, with the original predictive model being externally validated in each nation. By having nearly identical papers for each nation, it makes it easy to compare the results. This could be viewed as one large project, broken into separate papers by nation and QCovid algorithm version.

Minor comments:

1) What does “recalibrated” mean in Figure 2? How was this done?

RESPONSE: We have now clarified that recalibration involves rescaling multiplying predicted risk by a constant so that the expected total number of events is equal to the actual total number of events. We have amended the Model validation section to explain this more clearly (lines 248 – 253):

'We also calculated recalibrated risk by scaling predicted risks by a multiplicative constant so that expected total number of events predicted was equal to observed total number of events. We used this recalibrated risk to calculate Brier scores and made re-calibrated observed versus expected plots'

2) It would be helpful in the discussion to comment on whether the authors have any hypotheses on why the model overestimated risk so much. In particular, the text states that the way they handled missing data would have “had the effect of under-estimating risk”. As a result, the predicted risk might have been even higher if the data were available. Were the patient populations different in some way between Scotland and England that could explain this?

RESPONSE: We considered this at length, but do not have any plausible hypotheses. As you note, the differences in missing data between Scotland and England would intuitively lead to an expectation of risk being under-estimated rather than over-estimated. We are not aware of any significant differences between the populations of Scotland and England that would explain the discrepancy.

3) Why were there missing variables in Scotland? Was the model created for England without considering what data are available in Scotland and Wales? Are there any lessons learned from this study about what variables the UK nations have in common and whether future models should or should not focus on those variables?

RESPONSE: The QCovid models were created by the QResearch team using their database at the request of England's Chief Medical Officer. There are differences in data availability between UK nations. Although some variables were not available in Scotland, most variables were available, including those that are the most important predictors of severe COVID-19 outcomes (age, deprivation, etc). It is not clear whether overall model performance would have been improved by limiting to a common set of variables available across all nations, or including all variables known to be strong predictors of severe COVID-19 outcomes and imputing any missing data.

We note the helpful suggestion about taking a UK-wide perspective right from the outset and have now mentioned this in the revised Discussion when considering suggestions for future work in the following sentence (lines 370-371):

For future research, taking UK-wide perspectives on data availability when developing risk prediction tools should be considered if these tools are to be used nationally.

Reviewer: 2
Dr. Esther Gathogo, Cerner Corporation

Comments to the Author:
A well written paper. I have a few points to feedback

1. In the dataset you use, did you exclude any patients that may have opted out of use of their data for research purposes? If not, include process that you used to exclude opt out patients.

RESPONSE: We had approvals for this project to use anonymised patient data without consent and so there was not a pathway for patients to opt out. This is now made clearer in our revised Methods in the following sentence (lines 269-270):

As we had approvals to use anonymised patient data without consent, there was no pathway for patients to opt out.

2. Why did you make the assumption of Type I diabetes = uncontrolled and Type II = controlled. Is there Scottish data on the incidence or prevalence of controlled/uncontrolled of each diabetes type that you could use instead for imputation?

RESPONSE: We attempted to extract patient HbA1c levels from GP records, but were not able to due to permissions for new data extracts being revoked while we carried out the study. We are not aware of any up-to-date data with widespread coverage in Scotland, which we had permissions to access, on incidence of controlled/uncontrolled diabetes by type, and therefore we used modal substitution from the QResearch dataset covering the population of England to impute whether diabetes was controlled or uncontrolled.

3. Did the database have Height/Weight that you could use to calculate BMI instead of imputation?

RESPONSE: We did not have height/weight data in EAVE II - only BMI was available from the GP records. Again, this was a permissions issue. We were only permitted to have grouped data such as BMI as an integer and not exact height nor weight.

4. Is there data on COVID-19 variants being phase II of COVID-19 pandemic? The variants may explain the differences in rates of hospitalisations or death.

RESPONSE: Since we were comparing the results of the model over the same time-period to England, the dominant variants were the same; this should therefore not have impacted on our findings.

5. The model doesn't accurately predict older people, is it possible to evaluate age adjusted co-morbidities eg Charlson comorbidity index which has been reported as a predictor for mortality rate and severe clinical outcome in the patients with COVID-19.

RESPONSE: While we agree that this may be an interesting modelling approach, our aim in this paper was narrower and more centred around evaluating an existing model. One way this might be done in future work would be to include interactions terms between age and comorbidities in the model. We have conveyed this suggestion to the team who developed the QCovid models for future modelling.

6. I see COPD in your model, do you have data on infective exacerbation of COPD? COPD is one of the most common reasons for hospitalisation.

RESPONSE: We do not have data on infective exacerbation of COPD in EAVE II.

We are grateful for this opportunity to revise our work in the light of this thoughtful and constructive feedback. We trust that these revisions are to your satisfaction and that the manuscript is now

suitable for publication. Please do not however hesitate to contact us if you require any further revisions or clarification.

VERSION 2 – REVIEW

REVIEWER	Weber, Griffin Harvard Medical School
REVIEW RETURNED	23-Sep-2023

GENERAL COMMENTS	The authors addressed the minor comments I made in my previous review.
--

REVIEWER	Gathogo, Esther Cerner Corporation, Population Health Management
REVIEW RETURNED	08-Sep-2023

GENERAL COMMENTS	Thank you for responding to review comments. Looking forward to future updates on the QCOVID risk prediction algorithms.
--